# Modeling the Influence of Fake Accounts on User Behavior and Information Diffusion in Online Social Networks

Sara G. Fahmy [1,*], Khaled M. Abdelgaber [1,2], Omar H. Karam [1] and Doaa S. Elzanfaly [1,3]

1   Department of Information Systems, Faculty of Informatics & Computer Science,
    The British University in Egypt, Cairo 11511, Egypt
2   Department of Physics & Engineering Mathematics, Faculty of Engineering-Mataria, Helwan University,
    Cairo 11511, Egypt
3   Department of Information Systems, Faculty of Computer & Artificial Intelligence, Helwan University,
    Cairo 11511, Egypt
*   Correspondence: sara.gamil@bue.edu.eg; Tel.: +20-01274511500

**Abstract:** The mechanisms of information diffusion in Online Social Networks (OSNs) have been studied extensively from various perspectives with some focus on identifying and modeling the role of heterogeneous nodes. However, none of these studies have considered the influence of fake accounts on human accounts and how this will affect the rumor diffusion process. This paper aims to present a new information diffusion model that characterizes the role of bots in the rumor diffusion process in OSNs. The proposed $SI_hI_bR$ model extends the classical SIR model by introducing two types of infected users with different infection rates: the users who are infected by human ($I_h$) accounts with a normal infection rate and the users who are infected by bot accounts ($I_b$) with a different diffusion rate that reflects the intent and steadiness of this type of account to spread the rumors. The influence of fake accounts on human accounts diffusion rate has been measured using the social impact theory, as it better reflects the deliberate behavior of bot accounts to spread a rumor to a large portion of the network by considering both the strength and the bias of the source node. The experiment results show that the accuracy of the $SI_hI_bR$ model outperforms the SIR model when simulating the rumor diffusion process in the existence of fake accounts. It has been concluded that fake accounts accelerate the rumor diffusion process as they impact many people in a short time.

**Keywords:** rumors; fake accounts; social networks; bots; SIR model

## 1. Introduction

Social networks are online platforms that connect individuals who have similar interests to communicate, generate and share content. Although social networks have a lot of advantages, one of their drawbacks is that they may mislead and drive public opinion with unverified content that can be easily spread among users in an uncontrolled fast manner [1,2]. The spreading of posts and sharing information through social networks is known as information diffusion [3,4]. One of the main factors that affects the diffusion rate is the reaction of users when receiving information from other accounts. This reaction varies from one user to another depending on the user's beliefs and intentions. Some users reject information, some reject spreading information, and others accept and spread information [3]. The behavior of users on social networks can be manipulated and influenced by fake accounts that are known as bots. In other words, bots are software programs that carry out pre-defined repetitive tasks. They act like humans but much faster because they are automated [5,6]. In addition, they can perform many tasks such as producing content while hiding their robotic identity [7]. In 2010, social bots were employed during the U.S. elections to influence the followers of some candidates by directing them to websites that contain fake news [6]. Misinformation diffusion has also occurred during COVID-19 and the U.S. elections in 2020 [8]. Incapsula reports concluded that in 2014,

"about 56% of the internet traffic was generated by bots", and in 2019, malicious bot traffic on the internet rose by 18.1%, and currently, they are almost one-quarter of all internet traffic [9]. Therefore, understanding the behavior of bots and how they affect the dynamics of information diffusion in social networks is essential to stop their role in diffusing rumors and fake news. The mechanisms of information diffusion on OSNs have been studied extensively from various perspectives and with various approaches. This includes the time-series approaches that model the dynamics of the diffusion process over time using mathematical expressions [10]. Among these models are the epidemic models [1,4,11], the stochastic model [12–14], and the progressive models [1,15,16]. Other approaches are the data-driven ones [10] where machine learning and deep learning algorithms are used to model the diffusion process based on the features that are extracted from the collected datasets [17,18]. Some of these studies focus on identifying and modeling the role of heterogeneous nodes in the OSNs, such as the most influential users [19] and super spreaders [20]. However, none of these studies have considered the influence of fake accounts on human accounts and how this will affect the rumor diffusion process. In measuring the influence power of different network users, most researchers are using the structural properties of the network such as centrality measures [21,22] , PageRank index [23], and a—centrality [24]. Although these measures are useful in most cases, they are less accurate in the case of bots. This is because bot accounts usually constitute a lower percentage of the whole accounts in the network with a sparse degree distribution [25] and the average centrality measures of normal users are greater than the centrality measures of bots [25]. On the other hand, there is a reasonable amount of research that analyzes the behavior of bots and how they influence the diffusion of fake news and rumors in OSNs [2,5,25–28]. Most of these studies rely on statistical analysis methods to compare the behavior of bots and humans in the diffusion process of specific events [6,27]. However, modeling the behavior of bots and how they influence the mechanisms of rumor diffusion in OSN is still in its beginnings with very few attempts, such as the work in [27]. This leads to the necessity of creating a proposed model that measures the influential behavior of bot accounts and considers their special nature. In this context, this paper aims to present an information diffusion model that characterizes the role of bots in the rumor diffusion process in OSNs. The proposed model extends the classical SIR by containing two types of infected users with different infection rates: the users who are infected by human accounts ($I_h$) with a normal infection rate $\lambda$ and the users who are infected by bot accounts ($I_b$) with a different diffusion rate $\theta$ that reflects the intent and steadiness of this type of account to spread the rumors. The intuition for this differentiation relies on the findings of [27,29], and [30], where it has been shown that there is a significant difference in the impact of bot accounts and human accounts on information diffusion. Bot accounts tend to have a long-term behavior [6] and a greater influence than that of human accounts regarding specific topics [8,25,31]. Moreover, the proposed model is introducing the use of the social impact theory [32] for measuring the influence of bots on human accounts. This is because, unlike other influence measures, the social impact theory considers both the strength and the bias of the source node. This will better reflect the deliberate behavior of bot accounts to spread a rumor to a large portion of the network. The proposed model has been evaluated using a rumor dataset collected from Twitter [5]. The least-squares fitting function is used to estimate the parameters of the model, which can accurately predict the rumor propagation process. The rest of this paper is organized as follows. Section 2 contains related work for information diffusion models and social influence. Section 3 contains a detailed explanation of the proposed $SI_hI_bR$ model. Section 4 contains simulation and numerical results. Section 5 represents the conclusion and finally the references list.

## 2. Related Work

Two major research areas are related to this study: modeling the dynamics of the information diffusion process and measuring the influence of the network users.

### 2.1. Information Diffusion Models

As stated in the introduction, the two main approaches for modeling the information diffusion process are time-based and data-driven. Although researchers started to adopt the data-driven approaches because of the rapid development of machine learning and deep learning and due to their high prediction accuracy [25,33], most research in this area uses epidemic models [34–36]. This is because epidemic models are straightforward, efficient, extendable, interpretable, and conforming to some real-world physic laws [10,37], whereas the performance of the machine learning and deep learning algorithms is unexplainable as it is dependent on the quality and the quantity of the extracted features from the collected datasets [10]. Epidemic models, as the elementary models of information diffusion, study the diffusion process by its analogy with the diffusion of epidemics, since they both have similar diffusion mechanisms. There are four main epidemic models: the Susceptible–Infected model (SI) [38,39], Susceptible–Infected–Susceptible model (SIS) [1,38], Susceptible–Infected–Recovered model (SIR) [4,38,40], and Susceptible–Infected–Recovered–Susceptible model (SIRS) [1,38]. The main differences between these models lie in their distinctive ways of detecting the status and the behavior of each node and how it affects the diffusion process. Among these models are the SIR model and its extensions, which are the most commonly used ones specifically for rumor diffusion [33–36,41,42]. The SIR model has been reformed and extended in many types of research to incorporate different factors that affect the rumor diffusion process. In fact, these extensions can be categorized into three main categories based on the type of modification that has been applied to the basic SIR model. As shown in Tables 1–3, the first category contains the models that extend the SIR model by adding one or more new nodes to represent different states; the second category comprises the models that modify the classical SIR model by considering new factors that may affect the diffusion process, and the third category has the models that modify the classical SIR model by branching some of the main nodes to sub-nodes.

### 2.1.1. Classical SIR Model Extensions with New Node

This section explains the various extensions that have been applied to the classical SIR model by adding new types of users to represent different states for better diffusion results. For instance, previous studies added a hibernators node, an anti-rumor node, an exposed node, a potential spreader, a collector node, a hesitating node, adding a commentor node, and adding a lurker node in [17,18,33,35,42–45], respectively.

The SIHR model in [43] divides the social media users into four groups: "spreaders", "ignorant", "hibernators", and "removed". The spreader is the same as the infective state; ignorant is the same as the susceptible state. Stiflers are those who are aware of the news but do not publish it, which is similar to the removed state. The hibernator node represents how frequently information diffuses. In addition to the SKIR model, [18] reflects on the effect of the competition between rumor and anti-rumor information on the propagation process. They also considered the user behavior driving force by using evolutionary game theory and multiple information regression methods. Furthermore, the SEIR model in [17] inserts an exposed (E) node into the classical SIR model. Exposed nodes indicate infected individuals, but they are not yet able to infect others. In the SEIR model, the epidemic will not outbreak on the network unless the recovery number is more than an epidemic threshold. Otherwise, a great number of nodes in the network will become infected. This model constructs a dynamical evolution equation to properly define the information diffusion process. Moreover, it studies the impact of individual login rates and the number of friends on information diffusion. They concluded that individual login rates directly affect the information diffusion process. When a susceptible individual contacts infected individuals, the susceptible individuals transform into exposed ones with probability P. Meanwhile, $\epsilon$ represents the velocity for an exposed individual who will become an infected individual, and $\gamma$ represents the velocity for infected individuals who will be recovered individuals. Concerning the SCIR model in [46], it has has four states: "susceptible", "infected", "counterattack", and "recovered". The susceptible, infected and recovered nodes are the same nodes in the SIR model. The counterattack node is the node

that does not disseminate information and published the counter information. Once a susceptible individual interacts with an infected individual, the susceptible one may have three results: first, it turns out to be an infective node with probability $\alpha$ (spreading rate); second, it turns out to be a refractory node with probability $\beta$ (ignoring rate); third, it turns out to be a counterattack node with probability $\theta$ (refuting rate).

Unlike the above models which have four states, the SEIRS-C model in [41] has five states at any time: the susceptible state (S), the exposed state (E) represents individuals who believe the rumor and have the intention to spread it, the infected state (I), the counterattack state (C) represents the ones who heard the rumor and published the truth, and the recovered state (R). Regarding the SPIR model in [44], it introduced the idea of a potential spreader set. The equations of the SIR model could suffer from repeated calculations that will be far from the actual values if the traditional SIR model was solved discretely. To overcome this problem, the SPIR model introduced the idea of a potential spreader set, which is defined as a susceptible individual whose friends have at least one infected individual. Concerning the IRCSS model in [35], it considers sharing, reviewing, collecting, and stifling. Once an ignorant user interacts with a sharer, he/she transforms into one of the four possible statuses: a "reviewer" once he/she comments on the information with probability a, a "sharer" when she/he shares the information with probability ß, a "collector" if he/ she considers the collection value of the information with probability $\gamma$ and "stiffer" if he/she has no response on the rumor with probability $\theta$. Unlike the above models, the SEIR1R2 model in [34] added a new node, which is the exposed node, and divided the recovered nodes into two groups, which are R1 and R2. This model takes into consideration the influence of the rumor self-purification mechanism. SEIR1R2 divided people into five datasets: the susceptible people (S), the exposed people (E) who are the ones who believe and have the intention to spread the rumor, the infected people (I), the recovered1 people (R1)—those who recovered after being infected—and the recovered2 people (R2): those who recovered due to comments published by individuals (error correction. Regarding the ICST model in [42], it has main four nodes: the ignorant, the communicator, the sharer, and the stiffer. Once an ignorant deal with a sharer, there would be three possible results: the ignorant may comment on the rumor with rate $\alpha$ (the commenting rate); the ignorant may believe the rumor and share it with rate $\beta$ (sharing rate), and the ignorant may not be interested in the rumor with rate $\lambda$ (stifling rate). In addition, the commentor may share the information with probability $\eta$, and the sharer may lose interest to become stifler with probability $\delta$. The newly registered accounts have a growth rate of $\rho$. Independent spreaders are considered in the SIR model in [47]. Through the process of information diffusion, at every time step, any ignorant node can be converted to an independent spreader with a specific probability. An independent spreader represents the fact that users can obtain misinformation from other channels rather than from their neighbors within the network. The density of the independent spreaders can be calculated by two aspects: the misinformation attractiveness and its current overall popularity. Normally, the popularity can be represented by the density of the publisher.

Concerning the ILSR model in [45], it considers two different users (important users and ordinary users) based on the degree of each node. ILSR has four main statuses: the ignorant, the lurker (heard the rumor but temporarily not publishing it), the spreader, and recovered. When the ignorant deals with a spreader, the ignorant will transform to be a lurker or spreader with probability $\alpha1$ or $\alpha2$, respectively. Once the lurkers deal with spreaders, he/she will transform to be a spreader with probability $\beta$. As a spreader deals with a recovered, the spreader will transform to be recovered with probability $\delta$. In addition, this model divided users into two groups: important users (those who do not believe rumors easily) and ordinary users (those who easily believe rumors) based on the node degree.

**Table 1.** Different modifications to the classical SIR model by adding new node types.

| Model | Description |
|---|---|
| SIHR | Extended the classical SIR model by adding a group called **hibernators** to model the forgetting and remembering mechanisms of the infected accounts [43]. |
| SKIR | Studied the effect of the competition between the users who adopt the rumor and those who adopt an **anti-rumor** on the propagation process [18]. |
| SEIR | Added an **Exposed** node that represents people who are infected but not yet infectious [17]. |
| SICR | Introduced the **Counterattack** node to represent susceptible individuals who may not agree with the rumor [46]. |
| SEIRS-C | Updated the SICR in [46] by adding the **exposed** state along with the **Counterattack** state [41]. |
| SPIR | Adopted the concept of a **Potential Spreader set** of users to model the susceptible node that is likely to become infective at the next unit time [44]. |
| IRCSS | Added a **collector** state to model the value of the news [35]. |
| SEIR1R2 | Proposed a rumors purification model that contains **exposed** nodes and **two types of recovered** nodes: those who had never been exposed to the rumor or recovered from a rumor, and those who purified the rumor [34]. |
| SHAR | Added a **hesitating** state to include the users who heard the rumor, but they are uncertain whether to propagate the rumor or not [33]. |
| ICST | Added a **commentor** node to model the susceptible node that is likely to become infective at the next unit time [42]. |
| ILSR | Added a **lurker** node that heard the rumor but temporarily is not publishing it. In addition, they consider **two types of users** (important users and normal users) based on the degree of each node [45]. |

### 2.1.2. Classical SIR Model with the Consideration of New Factors

This section explains the previous modifications that have been added to the classical SIR model by considering different factors such as the reading rate and diffusion rate of the neighbor's behavior in [48,49], respectively. The Mb-RP model in [48] divides the population into three clusters: "the unknown" representing nodes that do not know the information, "the known" representing nodes that publish information and "the unconcern" representing nodes that published the information earlier and then lost interest in propagating it. Once users publish a tweet, their followers will see it with a specific probability. In other words, the probability is introduced by how many individuals read every tweet. As the followers read the tweet, they may retweet and spread it. Once the followers already read the information, the publisher will change to the unconcerned status with some speed until no publisher exists. Therefore, individuals pay attention to a topic for a specific period. Only a part of the spreader's followers can read the information published. The ISIR model is introduced in [37] demonstrates the transmission rate as a function of the infected nodes concerning the crowding effect. The crowding effect is known when a huge number of infected nodes exists in a group, and the affected contacts between susceptible and infected nodes do not rise fast. In the ISIR model, the infection ratio is not a static value but a function of the infected nodes number. The Fractional SIR model is presented in [49]. Once a user is flooded with a lot of news, the news may not be well diffused. The FSIR model is introduced to reflect the impact of neighbors on a user in the information propagation process. Because the entire amount of attention an individual can pay to social media is limited, it is assumed that the amount of impact from every neighbor is inversely proportional to the total number of friends that the individual has. Regarding the irSIR model in [40], it shows the adoption and abandonment of individual opinions by inserting an infection recovery kinetics process. Everyone who enters the network is expected to continue forever and then eventually lose interest as their friends lose interest. Hence, modifying the traditional SIR model is very important to contain the infectious recovery process which provides a superior description of online social networks.

2.1.3. Classical SIR Model with Divided Nodes

This section explains the modifications that have been made to the classical SIR model by dividing one node into two nodes such as the recovered node that can be divided into Ra and Ru, infected accounts can be divided into opinion leaders and normal users, infected accounts can be divided into super spreaders and ordinary spreaders or infected accounts can be divided into new and old spreaders in [20,36,50,51], respectively. The SIRaRu model in [50] extends the classical SIR model by dividing the removed state into two classes: Ra which represents the individuals who accepted the rumor and then lost interest to publish it and Ru which shows the individuals who rejected the rumor from the beginning. The OL-SFI model in [20] studies the impact of opinion leaders on different stages of information propagation within a short duration. The model differentiates between the contribution of opinion leaders and normal ones and builds upon these two nodes. Every user may be in one of the following four positions at any time: the susceptible state (S), the forwarding state which is influenced by opinion leaders (FL), the forwarding state which is influenced by normal users (FN), and the immune state (I). The SAIR model in [51] introduces super-spreaders to the classical SIR model. A super-spreader on OSNs would impact more individuals than ordinary individuals, making individuals influential to impact others. Therefore, the model introduced the SAIR model to describe the super-spreading phenomena in information diffusion. A in the model represents the super-spreader. The INSR model in [36] considers the age of infection. This model has four main nodes: "the ignorant", "the new spreader", "the old spreader", and "the stiffer". The relative time that has passed from the beginning of the infection is what differentiates new spreaders from old spreaders. There are few attempts in the literature that studied the role of bots in the information diffusion process in OSNs. Among these attempts is the one in [6] that presents a statistical analysis of the interplay between bots and information diffusion in two specific scenarios: manipulating public opinion in the American elections and publishing social spam campaigns in the tobacco-related conversation on Twitter. They concluded that bots generate more engagement than humans when retweeting politics-related information on Twitter. Another research study [27], models and simulates states of the diffusion of disinformation in social networks produced by bots. This model analyzes the behavioral patterns of the accounts involved in disinformation based on three assumptions: the delay in the start of the disinformation, the restricted number of bots that the disinformation agent can publish, and the limited capabilities of social networks to detect bots. In addition, the Kolmogorov–Smirnov statistic was implemented to check if the data follow a normal distribution, and it was concluded that the data did not follow a normal distribution. As stated in the above-mentioned studies, the SIR model is one of the most extensively used models to study rumor propagation in social networks in a variety of ways. However, none of these studies has incorporated the effect of the bot accounts in the diffusion models.

**Table 2.** Different Modifications to the Classical SIR Model by Considering New Factors.

| Model | Description |
| --- | --- |
| Mb-RP | Considered the**reading rate** as the susceptible node read the rumor many times, he/she may retweet and spread it [48]. |
| ISIR | Considered the **infection rate is not a static value**, but it differs according to the number of infected nodes [37]. |
| FSIR | Considered the **diffusion rate of the neighbor's behavior**. Once a person has a lot of information from neighbors, the information may not be well diffused [49]. |
| irSIR | Added **infection recovery process**. Each user who joins the network is expected to continue forever and then eventually lose interest as their friends lose interest [40]. |

**Table 3.** Different Modifications to the Classical SIR Model by Branching the Basic Nodes.

| Model | Description |
|---|---|
| SIRuRa | Divided the removed states into **Ra** (users accepted the rumor but lose interest) and **Ru** (users do not accept information at all) [50]. |
| OL-SFI | Studied the impact of opinion leaders as the **opinion leaders** will spread the news faster than the **normal ones** as they have many followers [20]. |
| SAIR | Introduced **super-spreaders** to the classical SIR model as they can impact more individuals than ordinary ones and make them influential to impact others [51]. |
| INSR | The age of infection is considered to have **new and old spreaders** [36]. |

*2.2. Measuring Social Influence*

Social influence is evident when a user or a group of users influences the behavior of other users. Measuring this influence to identify the most influential nodes and how they affect the information diffusion process is one of the ongoing research areas [52]. Among various measures that have been proposed in the literature are the centrality measures that are based on the structural properties of the network [12]. The basic well-known centrality measures are degree, betweenness, closeness, eigenvector, and Katz centrality measures. As the analytical usefulness of each measure depends mainly on the context of the network without considering the behavior or the intentions of the nodes who are defusing the rumors and how this will affect the propagation rate, some recent studies are combining multiple centrality measures in closed formulas instead of using each measure by its own [24,53,54]. None of these studies reflect the deliberated and biased behavior of social bots when posting a rumor. Social impact theory is another approach for quantifying the social force that changes the behavior of network users by considering different aspects [55]. The theory describes the social impact as a function of three forces: the strength or the power of the source, the immediacy or the proximity of the source, and the number of present sources in the network [32], as shown in Equation (1) [55]. When applying this to the OSNs, the strength is mapped to the number of friends (i.e., the degree of a node), the immediacy is mapped to the online distance between users (i.e., the number of edges in the shortest path connecting two vertices), and the number of people in the group is the total number of nodes in the OSNs as described in Table 4.

$$I_k = bN_k^{a-1} * \sum_{i,j} S_i / d_{i j}^2 \tag{1}$$

In Equation (1), N represents the number of people in the influence group. In the social impact model, there is the insertion of a means that the persuasion of a specific belief does not increase linearly with the number of people holding it. In addition, b represents the possibility of bias in expressing the belief and d represents the distance between the source and the target. The more people there are close to each other, the more impact they have on each other. Additionally, s represents the number of people who can see the post [55,56]. Other researchers are considering the two-way influence between the target and the source rather than the source only, and they name it dynamic social impact theory [57]. In this paper, we are adopting the one-way influence (from the bot as a source to its followers as the targets) as social bots will never be influenced by normal accounts.

**Table 4.** Symbol Description for $SI_hI_bR$ Model, Social Impact Theory, and Closeness Centrality.

| Symbol | Description |
| --- | --- |
| *s(t)* | The number of susceptible accounts at a specific time. |
| $i_h(t)$ | The infected real accounts. |
| $i_b(t)$ | The number of infected bot accounts. |
| *r(t)* | The number of recovered accounts. |
| $\lambda$ | The infection rate for real accounts. |
| $\theta$ | The infection rate for bot accounts. |
| $si_h$ | Susceptible accounts transform to be infected accounts due to the existence of infected real accounts. |
| $si_b$ | Susceptible accounts transform to be infected accounts due to the existence of infected bot accounts. |
| $\mu$ | The recovery rate. |
| $\mu i_h$ | The number of recovered accounts. |
| $I_k$ | The impact of a person on a target group. |
| *b* | The possibility of bias in expressing the belief. |
| *N* | The number of people in the influence group |
| *a* | This means that the persuasion of a specific belief does not increase linearly with the number of people holding it. |
| $S_i$ | The number of followers for an account. |
| $d_ij$ | The distance (number of hubs) between the source and the target. |
| *B* | The number of bot accounts (constant number). |

## 3. The $SI_hI_bR$ Proposed Model

The classical SIR model and all its extensions that simulate the diffusion process are assuming that all accounts are real ones without considering the existence of fake accounts (social bots). However, these fake accounts will never change their state; they will remain infected and will never be susceptible or recover nodes. Their effect appears in the diffusion rate. Therefore, the proposed model ($SI_hI_bR$) extends the classical SIR model by considering fake accounts, as well as the real ones, and modeling their effect on the rumor diffusion process. As shown in Figure 1, a new state has been added to the model to address the difference in behaviors between bots and ordinary users (aforementioned facts stated in [25,29,30]). This will allow us to vary the transition rates between states. We mainly focus on the impact of bots on forwarding rumors. The main reason is that although bots, as a special group of spreaders, are a minority, they have the ability to extremely prompt information propagation due to the automated nature of their activity and persuasiveness.

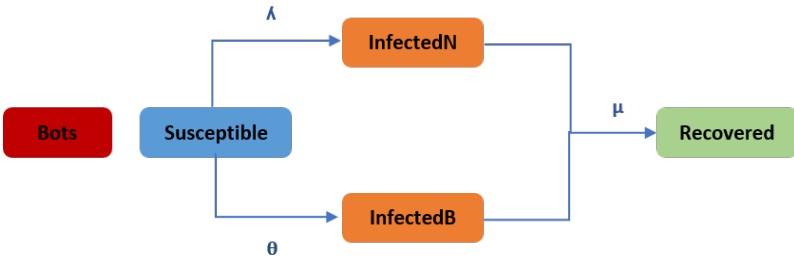

**Figure 1.** $SI_hI_bR$ Rumor Diffusion Model.

The proposed model has five main nodes which are the Bot node, the susceptible (*s*) node (the node that has not heard the news yet, but it can be infected in the future), the InfectedN ($i_h$) node (infected by normal accounts means this node knows the news and it is able to publish it), the InfectedB ($i_b$) node (infected by bot accounts), and the recovered (*r*) node (meaning that the node lost interest in the news). When a susceptible node contacts a bot node, the susceptible node may be transformed to be an InfectedB node by probability $\theta$, which is the bot infection rate calculated using social impact theory. If a susceptible node is contacted by a normal infected node, it may be transformed to be infectedN with probability $\lambda$, which is the normal infection rate. Then, all infected nodes will recover (except the bot accounts will not recover at any time) with rate $\mu$, which is the recovery rate. Considering the $SI_hI_bR$ rumor-spreading mechanism, the mean-field equations can be described as follows:

$$s(t) + i_h(t) + i_b(t) + B + r(t) = 1 \tag{2}$$

$$ds/dt = -\lambda s(t)i_h(t) - \theta s(t)i_b(t) \tag{3}$$

$$di/ds = \lambda s(t)i_h(t) + \theta s(t)i_b(t) - \mu i(t) \tag{4a}$$

$$di_h/dt = \lambda s(t)i_h(t) - \mu i_h(t) \tag{4b}$$

$$di_b/dt = \theta s(t)i_b(t) - \mu i_b(t) \tag{4c}$$

$$dr/dt = \mu(i_h(t) + i_b(t)) \tag{5}$$

**The $SI_hI_bR$ model relies on the following hypotheses:**

- Users are in a closed environment, meaning that the number of users (N) remains unchanged.
- The total population is divided into five groups: the susceptible nodes (s), the normal infected nodes ($i_h$), the bot-infected nodes ($i_b$), the bot nodes (B), and the recovered nodes (r).
- Each user, except bots, may be in one of four states at any given time: the susceptible state (S), the infected state influenced by bots ($I_b$), the infected state influenced by human users ($I_h$), and the recovered state (R).
- Correspondingly, if a susceptible node (S) contacts an infective node, the susceptible node will become infective with probability $\lambda$ if the infected node is a human account ($I_h$) or with probability $\theta$ if the infected node is a bot account ($I_b$).
- Bot accounts do not change their status. In other words, they will remain infected during one rumor propagation.
- Both infected types of nodes will recover with μ rate.
- It is not assumed that the user knows the identity of the account (fake or real) that posts the information, as fake accounts usually hide their identity. The effect of the fake accounts is reflected in the diffusion rate that differs from the normal rate. This is because social bots have a deliberate and continuous intent to post the rumor. Normal accounts post the information only once, whereas fake accounts keep posting the rumors for a while until it diffuses.

For measuring the $\theta$ (the infection rate of bot accounts), we are using Equation (1) [55], which depicts the social influence of bot accounts using the social impact theory.

$$\theta = bN_k^{a-1} * \sum_{i,j} S_i/d_ij^2 \tag{6}$$

**where:**

- *S*: The strength is mapped to the number of accounts that can see the rumor.
- *d*: The immediacy is mapped to the shortest distance between the bot account and the susceptible accounts.

- $N$: represents the number of accounts in the influence group.
- $a$: means that the persuasion of a specific belief does not increase linearly with the number of bots holding it.
- $b$: represents the possibility of the bias of bot accounts in expressing the belief.

The distance, $d_{(i,j)} = C(i)$, is calculated using closeness centrality in Equation (7) [21]. Closeness centrality calculates the shortest distance between the source and the target (it could be calculated from the network).

$$C(i) = ((n-1)/(\sum_{j=1}^{n} d_{ij})) \tag{7}$$

There are a lot of numerical methods to solve ordinary differential equations (ODE); one of them is the Runge–Kutta method. Therefore, the system from Equation (3)–(5) can be resolved using the Runge–Kutta method (another form of solution for the differential equations) in the vector form as:

$$Z'(t) = F(t, Z(t)), \quad 0 \le t \le t_{max} \tag{8}$$

$$Z(t) = [(s(t) \quad i_h(t) \quad i_b(t) \quad r(t))] \tag{9}$$

$$\hat{p} = p + 1 \tag{10}$$

where vector Equation (9) and $t_{max}$ is used as an adaptation for $t \to \infty$. To obtain local error estimation for adaptive step-size control effectively, consider two Runge–Kutta formulas of different orders p and Equation (10). A Runge–Kutta process generates a sequence $Z_n$ as an approximation of $Z(t_n)$ for $0 = t_0 < t_1 < ... < t_n = t_{max}$. In the interval from $t_n$ to $t_{(n+1)} = t_n + h$, there are two approximations of $Z(t_{(n+1)})$ called $Z_{(n+1)}$ and $\hat{Z}_{(n+1)}$ for $p$ and $\hat{p} = p + 1$, respectively. Their forms are the following:

$$Z_{(n+1)} = Z_n + h_n \sum_{i=0}^{m} b_i K_i \quad and \quad \hat{Z}_{(n+1)} = Z_n + h_n \sum_{i=0}^{m} \hat{b}_i K_i \tag{11}$$

where $m$ is the number of stages and

$$K_0 = F(t_n, Z_n), K_i = F(t_n + c_i h_n, Z_n + h_n \sum_{j} = 0^{i-1} a_{ij} K_j), \quad and \quad i = 1, 2, \ldots, m. \tag{12}$$

By taking $p = 4$ and $m = 7$, the coefficients $a_{ij}, b_i, \hat{b}_i$, and $c_i = \sum_{j=0}^{i-1} a_{ij}$ where $i = 1, 2, \ldots, m$ can be evaluated as shown by Bogacki and Shampine [58] to produce the efficient pair of formulas $Z_{(n+1)}$ (4th order formula) and $\hat{Z}_{(n+1)}$ (5th order formula). The error between the two numerical solutions $Z_{(n+1)}$ and $\hat{Z}_{(n+1)}$ is calculated by

$$e_{(n+1)} = max|Z_{(n+1)} - \hat{Z}_{(n+1)}| \tag{13}$$

In case of $e_{(n+1)} = t$, one can use $Z_{(n+1)}$ or $\hat{Z}_{(n+1)}$ as the final approximate value of $Z(t_{(n+1)})$ where $t$ is the required accuracy. On the other hand, if $e_{(n+1)} > t$, the error $e_{(n+1)}$ is used to adapt the step size $h_n$ to $\hat{h}_n$ as follows [59]

$$\hat{h}_n = h_n (\tau/(e_{(n+1)}))^{(1/p)} \tag{14}$$

The adapted step size is used to estimate the new values of $Z_{(n+1)}$ and $\hat{Z}_{(n+1)}$ until achieving $e_{(n+1)} = \tau$.

## 4. Simulation and Numerical Results

While the previous section explains the $SI_hI_bR$ model and how it will be implemented, this section will consider the impact of fake accounts on information diffusion using numerical simulation.

### 4.1. Experimental Design

The model has been evaluated through two sets of experiments. In the first set, the diffusion process has been simulated without the consideration of the existence of bots (using the Classical SIR model). In the second set, the simulation has been performed with the consideration of bot accounts using the $SI_hI_bR$ model. The two models have been evaluated against a real dataset obtained from Twitter [5]. The dataset contains 4929 accounts: 1455 identified as bot accounts and 3474 identified as human accounts. Additionally, it contains the tweet's text, time, account name, ID, followers, friends, and tweet ID. It covers the tweets of real and bot accounts across different years.

For better tracking of the rumor diffusion, we extracted the tweets published by bots and human accounts in the same period across two years to have two sub-datasets, one containing the 2011 tweets and the other containing the 2012 tweets. In addition to the tweets, the retweets and the replays are considered infected users. In the 2012 dataset, the initial number of infected accounts is 53 accounts (the number of real accounts is 16 and the number of bots is 37), and the total number of infected accounts is 11,219. As in the 2011 dataset, the initial number of infected accounts is 13 accounts (nine real accounts and four bots), and the total number of infected accounts is 2300.

Figures 2–5 represent the simulation results of the SIR and the $SI_hI_bR$ in the 2012 and 2011 datasets. Different infection and recovery rates are tested as shown in Tables 5 and 6 to decide the best-fit rates with respect to the two datasets. It is obvious from the figures that the infection and recovery rates control the model; therefore, different rates are used. Figures 2 and 3 show the results of implementing the classical SIR model across 2012 and 2011 datasets using $\lambda = 0.3$, $\mu = 0.1$ and neglecting $\theta$ (the best-fit parameters). On the other hand, Figures 4 and 5 represent the results of implementing the proposed model ($SI_hI_bR$) over 2012 (using $\lambda = 0.3$, $\mu = 0.1$ and $\theta = 1.38$) and over 2011 datasets (using $\lambda = 0.3$, $\mu = 0.1$ and $\theta = 0.48$).

### 4.2. Testing the SIR Model

The classical SIR model has been tested against the 2011 and 2012 datasets with three different rates of infection and recovery. This simulation of the 2012 and 2011 datasets will disregard the bot accounts (bot number = 0). Figures 2 and 3 reflect the normal behavior of the SIR model where the number of infected accounts increases while the susceptible accounts decrease and then the recovered accounts begin to appear. The simulation terminates when the number of infected accounts becomes 0.

Table 5 represents the results of the simulation using different infection and recovery rates to be compared with the real dataset values. From simulation results, the maximum number of infected accounts and the time when they are infected can be noticed. However, the exact time of reaching the maximum number of infected accounts cannot be concluded from the dataset, as it was published for bot identification and not for diffusion simulation. Therefore, to be able to obtain the number of infected accounts from this dataset, the number of tweets, retweets, and replies is considered. The exact time for retweets and replies is not mentioned in the dataset. As shown in Table 5, the best-fit rates are achieved when using an infection rate $\lambda$ of 0.3 and a recovery rate $\mu$ of 0.1.

**Table 5.** The SIR Model Simulation Results.

| Year | Initial Infected | $\lambda$ | $\mu$ | Total Number of Infected Accounts | Days |
|------|-----------------|-----------|-------|-----------------------------------|------|
| 2011 | 9 Real Accounts<br>4 Bot Accounts | 0.3 | 0.1 | 704 | 30 |
|      |  | 0.4 | 0.2 | 365 | 30 |
|      |  | 0.8 | 0.3 | 604 | 12 |
| 2012 | 16 Real Accounts<br>37 Bot Accounts | 0.3 | 0.1 | 3448 | 30 |
|      |  | 0.4 | 0.2 | 1786 | 31 |
|      |  | 0.8 | 0.3 | 2959 | 12 |

It can be concluded that the classical SIR model is far from reality, as it neglects the bot accounts and the fact that they remain infected and never recover. It does not reflect the diffusion process of the real datasets. In the 2011 and 2012 datasets, the total number of infected accounts was 2300 and 11,219, respectively, whereas the SIR model simulation gives 704 for 2011 and 3448 for 2012, which represents about 31% of the real number of infected accounts. These numbers have been achieved with the best-fit rates.

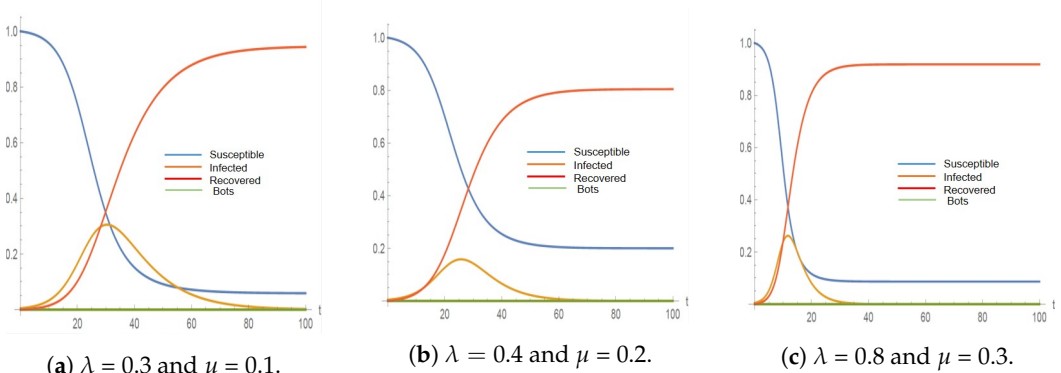

(**a**) $\lambda = 0.3$ and $\mu = 0.1$.  (**b**) $\lambda = 0.4$ and $\mu = 0.2$.  (**c**) $\lambda = 0.8$ and $\mu = 0.3$.

**Figure 2.** The simulation results for SIR model using different rates on the 2012 dataset (a is the best fit).

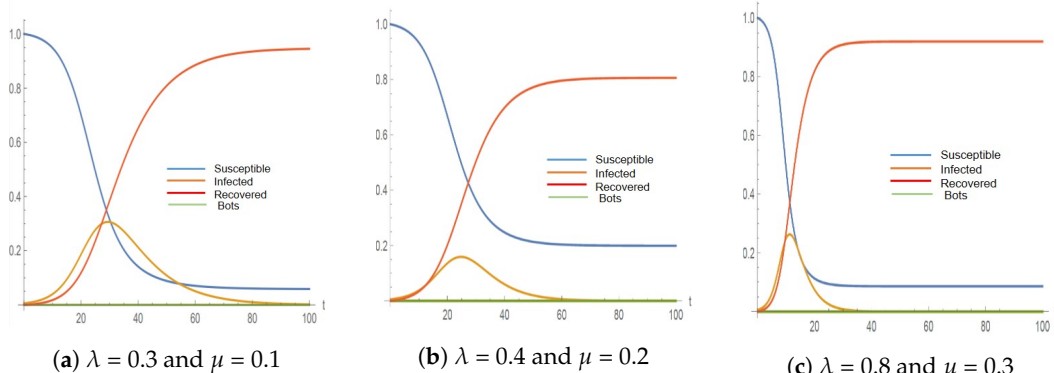

(**a**) $\lambda = 0.3$ and $\mu = 0.1$  (**b**) $\lambda = 0.4$ and $\mu = 0.2$  (**c**) $\lambda = 0.8$ and $\mu = 0.3$

**Figure 3.** The simulation results for SIR model using different rates on the 2011 dataset (a is the best fit).

*4.3. Testing the $SI_h I_b R$ Model*

The $SI_h I_b R$ model has been tested against the 2011 and 2012 datasets with three different rates of infection and recovery. While the bot infection rate $\theta$ is constant as it is calculated using social impact theory, this simulation will consider the bot accounts and their bias, as shown in Figures 4 and 5. In this stage, a variation will happen between infected human accounts and bot accounts. The simulation reflects the $SI_h I_b R$ model where the number of infected accounts (due to the existence of bot accounts and human accounts) increases while the susceptible accounts decrease and then the recovered accounts begin to appear. The simulation terminates when the number of infected accounts is equal to the

number of bot accounts. This means that they will not recover at any time and they will remain infected.

Table 6, represents the results of simulating the $SI_hI_bR$ model in comparison to the real dataset.

Figure 6 and 7 show how the number of infected nodes changes by using different infection and recovery rates ($\lambda$ and $\mu$) across the classical SIR model and the $SI_hI_bR$ model.

When comparing the simulation results with the 2011 and 2012 datasets, we found that the best fit is achieved for the $SI_hI_bR$ model when setting the infection rate to $\lambda = 0.3$ and the recovery rate to $\mu = 0.1$.

The $SI_hI_bR$ better matches the real datasets. The simulation gives 1032 infected accounts in the 2011 dataset and 8359 in the 2012 dataset. This represents about 44% of the 2011 dataset and 74.5% of the infected accounts in the 2012 dataset, which is close to reality. Moreover, the influence of fake accounts on the diffusion process is reflected in the number of days when the infection reaches its peak. In Table 5, without considering the bots in the SIR model, the infection outbreak is on day 30, whereas in Table 6, the rumor reaches its outbreak on day 20 in the 2011 dataset and on day 7 when using the 2012 dataset.

Comparing the classical SIR model with the new $SI_hI_bR$ model on the 2012 dataset, it can be noticed that in the SIR model, the maximum number of infected accounts is 3448 (reached after 30 days) as shown in Figure 2, while in the $SI_hI_bR$ model, the maximum number of infected accounts is 8359 (reached after 7 days), as shown in Figure 4. When comparing the two models in the 2011 dataset, it is observed that in the classical SIR model, the maximum number of infected accounts is 704 (reached after 30 days), as shown in Figure 3, while in the $SI_hI_bR$ model, the maximum number of infected accounts is 1032 (reached after 20 days), as shown in Figure 5.

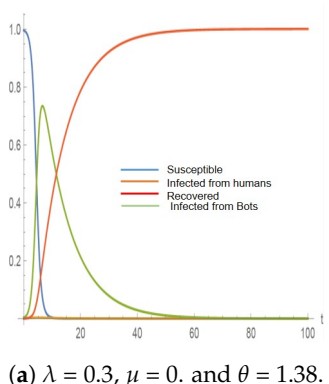

(**a**) $\lambda = 0.3$, $\mu = 0$. and $\theta = 1.38$.

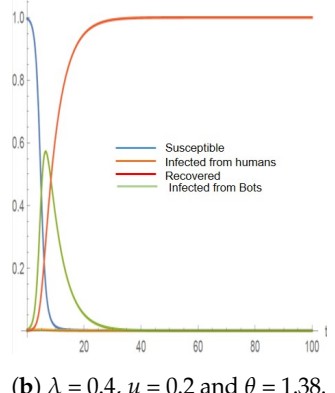

(**b**) $\lambda = 0.4$, $\mu = 0.2$ and $\theta = 1.38$.

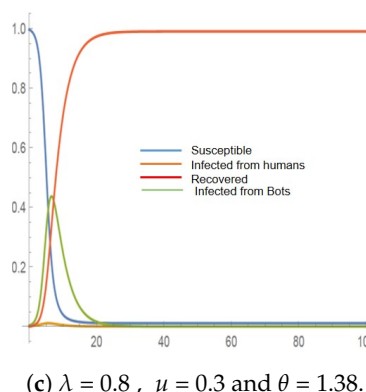

(**c**) $\lambda = 0.8$ , $\mu = 0.3$ and $\theta = 1.38$.

**Figure 4.** The simulation results for the $SI_hI_bR$ model using different rates on the 2012 dataset (a is the best fit).

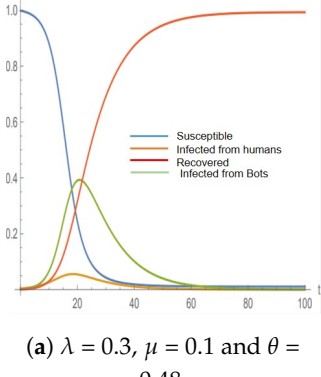

(**a**) $\lambda = 0.3$, $\mu = 0.1$ and $\theta = 0.48$.

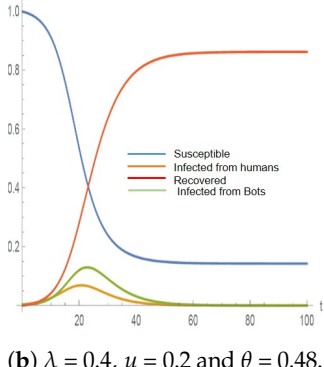

(**b**) $\lambda = 0.4$, $\mu = 0.2$ and $\theta = 0.48$.

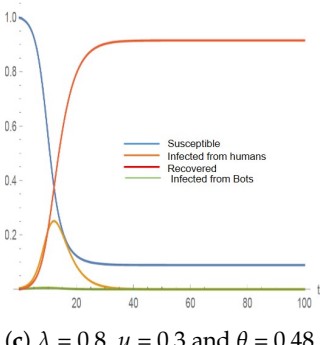

(**c**) $\lambda = 0.8$, $\mu = 0.3$ and $\theta = 0.48$.

**Figure 5.** The simulation results for the $SI_hI_bR$ model using different rates on the 2011 dataset (a is the best fit).

**Table 6.** The $SI_hI_bR$ Model Simulation Results.

| Year | Initial Infected | $\lambda$ | $\mu$ | $\theta$ | Num of $I_b$ | Num of $I_h$ | Total Num of I | Days |
|------|------------------|-----------|-------|----------|--------------|--------------|----------------|------|
| 2011 | 9 Real Accounts<br>4 Bot Accounts | 0.3 | 0.1 | 0.48 | 904 | 128 | 1032 | 20 |
|      |                  | 0.4 | 0.2 | 0.48 | 298 | 158 | 456 | 22 |
|      |                  | 0.8 | 0.3 | 0.48 | 13 | 576 | 589 | 12 |
| 2012 | 16 Real Accounts<br>37 Bot Accounts | 0.3 | 0.1 | 1.38 | 8306 | 33 | 8359 | 7 |
|      |                  | 0.4 | 0.2 | 1.38 | 6467 | 32 | 6499 | 7 |
|      |                  | 0.8 | 0.3 | 1.38 | 4937 | 126 | 5063 | 6.5 |

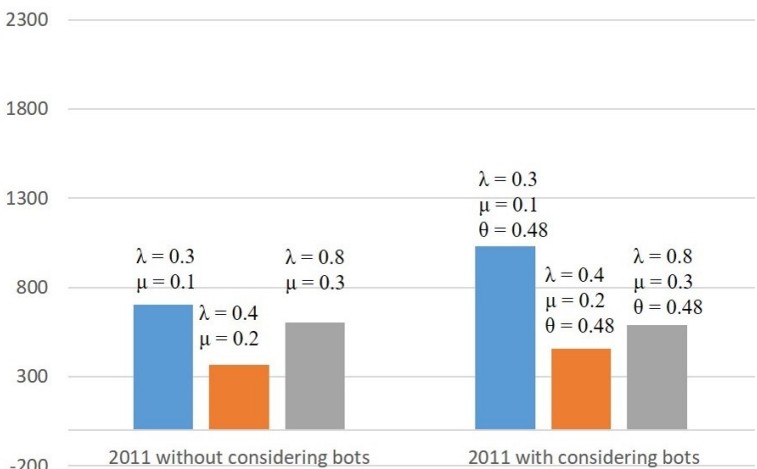

**Figure 6.** Number of infected accounts comparison for 2011.

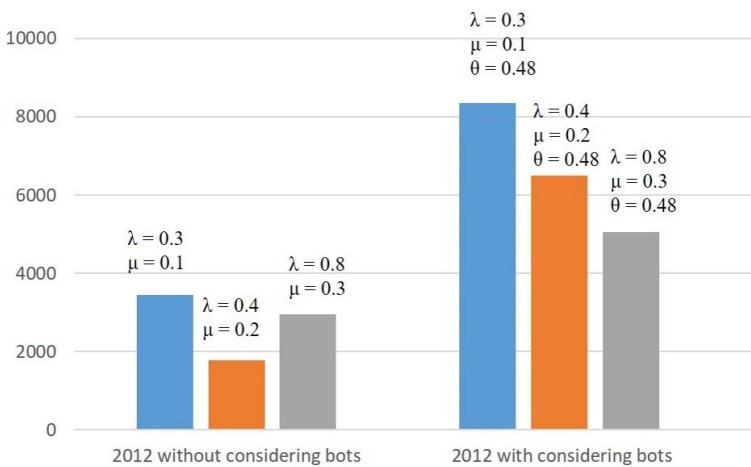

**Figure 7.** Number of infected accounts comparison for 2012.

It is worth mentioning that when the infection rate of the human accounts $\lambda$ exceeds the infection rate of the bot accounts $\theta$, the infection outbreak did not change in the two models in the 2011 dataset (on day 12) as the human accounts are more infectious than the bots. However, it changes in the 2012 dataset as the number of bots is more than the number of human accounts.

As a result, the $SI_hI_bR$ model reflects the effect of deliberate and biased behavior on real accounts, because fake accounts persuade more people in a shorter period of time. Unlike normal accounts, fake accounts can impact more accounts and make them influential enough to impact others. Therefore, increasing the number of fake accounts speeds up

the rumor propagation process. Even if the number of fake accounts is small, they can have a strong influence on others. Finally, we can say that the $SI_hI_bR$ model works more effectively than the classical SIR model in the presence of bot accounts.

## 5. Conclusions

This research considers the effect of bot accounts on rumor propagation in social media. Two types of nodes have been introduced, by including both human accounts and bot accounts. In addition, a new infection rate for the bot accounts is introduced using social impact theory that studies the impact of bot accounts on the real accounts. The analysis has been made on the results of both the classical SIR model and the $SI_hI_bR$ model. The two models have both been applied to the simulator. The results are then compared to a real dataset that contains real accounts tweets and bot tweets in 2011 and 2012. The results of the proposed model represent an improved accuracy compared to the classical model. The accuracy for 2011 without considering the effect of bots is 30.6%, and the accuracy considering the bot accounts is 44.8%. The accuracy for 2012 without considering the effect of bots is 30.7%, and the accuracy considering the bot accounts is 74.3%. Results point out the importance of using the $SI_hI_bR$ model in modeling rumors propagation, as bots can impact many accounts in a shorter time. Therefore, the $SI_hI_bR$ model proves that it is better than the classical SIR model. Future work is required to study the characteristics that affect the diffusion process to find the exact ratios between infection rates and recovery rates.

**Author Contributions:** S.G.F.: Roles Conceptualization, Data Curation, Formal Analysis, Investigation, Methodology, Resources, Software, Visualization, Writing—Original Draft; K.M.A.: Formal Analysis, Writing—Review and Editing; O.H.K.: Supervision, Writing—Review and Editing; D.S.E.: Roles Conceptualization, Data Curation, Methodology, Formal Analysis, Supervision, Validation, Visualization, Writing—Review and Editing. All authors have read and agreed to the published version of the manuscript.

**Funding:** This research received no external funding.

**Data Availability Statement:** The dataset was retrieved from http://mib.projects.iit.cnr.it/dataset.html, accessed on 1 November 2021.

**Conflicts of Interest:** The authors declare no conflict of interest.

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
