# Peer review of "Modeling the Influence of Fake Accounts on User Behavior and Information Diffusion in Online Social Networks"

_informatics, doi:10.3390/informatics10010027_

Round 1

Reviewer 1 Report

This paper proposed a new model named SI_hI_bR model which considers the influence of fake accounts in online social networks. However, I do not think this model is really “new”.  My main concerns are as follows:

1. In reality, it is usually difficult for a normal user to discern whether an account is fake or real. If he/she knows an account is fake, he/she will certainly not believe the contents it posted. Therefore, the model which introduces two different infection rates for real human beings and fake accounts seems not reasonable.

2. The analysis of the model is too simple. The authors only solved the differential equations and showed how the number of nodes with different states changes with time, which lacks a deep discussion. For example, what is the outbreak threshold of their model?  What is the influence of the number of fake accounts?

3. There are some typing errors in the manuscript. For example, the core equation in the part of model description (see equation 4a).

Therefore, I do not recommend this manuscript to be published.

Reviewer 2 Report

(1) In addition to references 6 and 7, is it possible to add cases by bots from the last two years? (2) I suggest that the author(s) increase the discussion of the results to enhance the research implications of the paper.

Reviewer 3 Report

This paper reviews SIR-type models for information diffusion, and presents a model for the same where bots are included as a means to spread a rumor. The authors simulate this model and provide some comparisons with a Twitter dataset, using Social Impact Theory.

The best part of the paper is the literature review of related information diffusion models, which is very comprehensive and well structured. Unfortunately, the presentation of the model is not nearly so clear, to the extent that I can't really understand what the model is enough to give a critical appraisal. The mathematics in equations (9) and underneath has typos and poor typesetting, and some terms are not defined. It's also not clear why this Runge-Kutta process needs to be described and used -- why not just apply a standard numerical ODE solver to the system (2-5)?

There is another typo in (2) -- what is B? -- and the description of how measuring \theta takes place is unclear: "we are using equation 1", but \theta does not appear in (1). Also, the distance d_{i,j} is said to be calculated using closeness centrality, but what is the network? This key point is not explained. Maybe it comes from the friend/follower lists described in Section 4.1, but it isn't clear.

Minor issues:

- inline mathematics needs to be italicized

- more typos in (7)

- figures 2-5 have small text, and are very similar. There isn't great discussion of the difference between these figures and why they were chosen to be included.

- there are many free parameters required in equation 1, which have some interpretation, but seem difficult to determine from experiment or data. The numerical investigations of the model go some way towards this, but some discussion of a possible inference framework (is there an optimization problem that could be solved to infer parameters?) would be useful if anyone wanted to implement this on another dataset.

Round 2

Reviewer 1 Report

The authors have answered my questions raised last. I recommend it to be published. There is one more suggestion: it is better to make a figure to show how the number of infected nodes changes with \lambda/\mu. 
